# The Blinking of Small-Angle X-ray Scattering Reveals the Degradation Process of Protein Crystals at Microsecond Timescale

**DOI:** 10.3390/ijms242316640

**Published:** 2023-11-23

**Authors:** Tatsuya Arai, Kazuhiro Mio, Hiroki Onoda, Leonard M. G. Chavas, Yasufumi Umena, Yuji C. Sasaki

**Affiliations:** 1Department of Advanced Materials Science, Graduate School of Frontier Sciences, The University of Tokyo, 5-1-5 Kashiwanoha, Kashiwa 277-8561, Chiba, Japan; ycsasaki@edu.k.u-tokyo.ac.jp; 2AIST-UTokyo Advanced Operando-Measurement Technology Open Innovation Laboratory (OPERANDO-OIL), National Institute of Advanced Industrial Science and Technology (AIST), 6-2-3 Kashiwanoha, Kashiwa 277-0882, Chiba, Japan; kazu.mio@aist.go.jp; 3Synchrotron Radiation Research Center, Nagoya University, Furo-Cho, Chikusa-Ku, Nagoya 464-8603, Aichi, Japan; h.onoda@nusr.nagoya-u.ac.jp (H.O.); l.chavas@nusr.nagoya-u.ac.jp (L.M.G.C.); 4Department of Applied Physics, Graduate School of Engineering, Nagoya University, Furo-Cho, Chikusa-Ku, Nagoya 464-8603, Aichi, Japan; 5Center for Synchrotron Radiation Research, Japan Synchrotron Radiation Research Institute, 1-1-1 Kouto, Sayo-cho 679-5198, Hyogo, Japan

**Keywords:** small-angle X-ray blinking (SAXB), time-resolved observations, protein crystal degradation, X-ray diffraction

## Abstract

X-ray crystallography has revolutionized our understanding of biological macromolecules by elucidating their three-dimensional structures. However, the use of X-rays in this technique raises concerns about potential damage to the protein crystals, which results in a quality degradation of the diffraction data even at very low temperatures. Since such damage can occur on the micro- to millisecond timescale, a development in its real-time measurement has been expected. Here, we introduce diffracted X-ray blinking (DXB), which was originally proposed as a method to analyze the intensity fluctuations of diffraction of crystalline particles, to small-angle X-ray scattering (SAXS) of a lysozyme single-crystal. This novel technique, called the small-angle X-ray blinking (SAXB) method, analyzes the fluctuation in SAXS intensity reflecting the domain fluctuation in the protein crystal caused by the X-ray irradiation, which could be correlated with the X-ray-induced damage on the crystal. There was no change in the protein crystal’s domain dynamics between the first and second X-ray exposures at 95K, each of which lasted 0.7 s. On the other hand, its dynamics at 295K increased remarkably. The SAXB method further showed a dramatic increase in domain fluctuations with an increasing dose of X-ray radiation, indicating the significance of this method.

## 1. Introduction

X-rays serve as indispensable quantum probes to investigate the three-dimensional structures of protein molecules, providing critical insights into their functions. However, the use of X-rays itself causes damage to proteins, compromising the accuracy of the acquired structural data. The X-ray-induced damage to protein crystals is a complex phenomenon, characterized by various structural alterations, radiation-induced chemical modifications, and conformational changes. Traditionally, efforts to assess X-ray-induced protein crystal damage have primarily relied on techniques such as wide-angle X-ray diffraction (WAXD) [1,2] and small-angle X-ray scattering (SAXS) [3,4,5]. The former assesses atomic-level structural deterioration, and the latter analyzes the dimensions of crystal domains larger than those observed with WAXD, while both methods generally focus on stable protein structures. However, the damage to protein crystals caused by X-rays can occur on the micro- to millisecond timescale, highlighting the need to measure such processes in real-time.

Diffracted X-ray Tracking (DXT) was proposed as a time-resolved observation method for the dynamics of biomolecules and particles including globular proteins [6], membrane proteins [7], coiled proteins [8], and supersaturated networks [9]. This method utilizes a polychromatic synchrotron X-ray beam to measure the real-time rotational motion of single biomolecules labeled with nanocrystals at the micro-second scale by tracking the movement of diffraction spots derived from the labels. A detailed explanation of this method is provided in the references [10]. The diffracted X-ray blinking (DXB) method was proposed as a monochromatic X-ray version of the DXT method [11]. DXB assesses the fluctuation in the diffracted X-ray intensity of nanocrystals labeled on protein molecules by using an autocorrelation function (ACF), which is directly correlated with the magnitude of the protein dynamics. Since the DXB method utilizes monochromatic X-ray sources including laboratory X-ray tubes [12], it has an advantage over DXT in terms of versatility and reduced damage, which enables us to monitor the dynamics of protein molecules on living cells [13]. In addition, this method can be applicable to the dynamics of crystalline objects themselves [14]. Remarkably, DXB can not only measure the diffracted X-rays of crystalline objects but also the X-ray haloes of non-crystalline ones, which enables us to assess the dynamics of polymeric materials [15,16]. Here, we examined a combined use of DXB and SAXS methods for a lysozyme single-crystal to obtain information about its domain dynamics caused by X-ray irradiation, which could be correlated with the X-ray induced damage. This novel combination technique, called the small-angle X-ray blinking (SAXB) method, would be applicable not only to evaluate the damage of protein crystals but also to visualize the vibration of the crystal domain at the micro- to millisecond timescale.

## 2. Results and Discussion

### 2.1. Concept of the Small Angle X-ray Blinking (SAXB) Method

Figure 1A,B show the experimental setup for the measurement of SAXB at the beamline BL2S1 of the Aichi Synchrotron Radiation Center. A lysozyme single-crystal mounted on a loop was exposed to a monochromatic X-ray beam, and its time-resolved X-ray diffraction and direct beam were recorded with a 2D-photon counting detector at a time resolution of 70 μs/frame. The sample temperature was controlled with LN gas flow. Figure 1C shows the converted time-averaged 1D X-ray scattering pattern for the SAXS-region of the lysozyme crystal. The magnitude of intensity fluctuations in small-angle scattering near the direct beam (show in Figure 1C) was assessed with the autocorrelation function at each pixel (Figure 1D), by which the fluctuation in domain distance was evaluated. This 1-pixel ACF curve (Figure 1D, cyan close circle) was fitted with a single exponential function (gray line, *ACF*(*t*) *= Aexp*(*−Γt*) + *y*), and its decay constant value (*Γ*) was evaluated. In the case of traditional DXB (Figure 1E), which measures the rotational motion of crystal from intensity fluctuations in its wide-angle X-ray diffraction, the size of the observed crystal is not limited as long as its diffraction can be observed (usually <20 nm). On the other hand, SAXB analyzes the intensity fluctuation in the SAXS region, and thus observes the dynamics of the crystal domain or the fluctuation in domain distance, which is derived from X-ray irradiation and can be correlated with the X-ray-induced damage of the crystal.

### 2.2. Temperature-Dependent X-ray Damages Measured with SAXB

We conducted two X-ray diffraction measurements at 95K and subsequently performed two more measurements at 295K on the same crystal. Figure 2A shows the time-averaged X-ray diffraction images of a lysozyme crystal. The number of diffraction spots, especially at the higher 2θ-angle, decreased with each measurement, indicating the occurrence of X-ray-induced damages. ACF analysis was performed for the SAXS region to extract information about the dynamics of domain fluctuation. Figure 2B is an average normalized ACF curve calculated from the intensity fluctuation in each pixel. As the damage to the crystal progresses, the ACF curve decays faster, suggesting the mobility of nano-domain structures increases. This result is also confirmed in Figure 2C,D, which shows the distribution of the decay constant values calculated from the single exponential fitting against the 1-pixel ACF curves, which is an indicator of the mobility of the target q-range (0.1–0.87 nm^−1^). When the crystal was exposed to the X-ray beam at 95 K, there was no difference between the decay constant distributions of the first and second measurements, whose median values were 4.224 s^−1^ and 4.067 s^−1^, respectively. This result suggests that dynamic motion of the crystalline domain was not significantly changed at this temperature, which indicates X-ray radiation damage did not affect the crystal. On the other hand, the decay constant value was significantly increased for the first measurement at 295 K (median = 5.011 s^−1^). This is indicative of the crystal domain dynamics increasing at this temperature, which could be attributed to the occurrence of X-ray radiation-induced damage. For the second measurement at 295 K, the decay constant value further increased (median = 6.454 s^−1^), which could only be ascribed to X-ray-induced damage. In this measurement, the distribution of the decay constant value became broadened, which suggests the heterogeneous dynamics of the crystal domain probably due to the co-existence of degraded and non-degraded regions. These results suggest that the dynamics of the crystal domain increase with the accumulation of damage and that such motion can be observed with the SAXB method.

### 2.3. Dose-Dependent X-ray Damages Measured with SAXB

To evaluate dose-dependent damage, we prepared another lysozyme crystal and conducted X-ray diffraction measurements twice at 295K on the same lysozyme crystal, followed by a 100 s X-ray irradiation, and then conducted two more X-ray diffraction measurements (Figure 3). As shown in Figure 3A, the number of X-ray diffraction spots derived from the lysozyme crystal decreased after the 100-second X-ray irradiation. Figure 3B shows an average normalized ACF curve calculated from the X-ray intensity fluctuation in this sample. As radiation dose of X-ray increased, the ACF curve decays faster, suggesting the mobility of nano-domain structures increases. As this experiment was performed at 295K, decay constant values of the first and second measurements were significantly different even before 100 s X-ray irradiation (median = 4.850 and 5.340 s^−1^ for the first and second measurements, respectively). After the 100 s X-ray irradiation, the decay constant value was dramatically increased (Figure 3C, median = 6.636 s^−1^), which could be ascribed to the X-ray-induced damage. In addition, the distribution of the decay constant value became broad for the first measurement, suggesting the existence of both slow and fast dynamics. The decay constant value of an additional measurement (second, after irradiation) further increased (median = 7.241 s^−1^), which suggests that the decrease in the slow dynamics resulted in the sharpening of the decay constant’s distribution. These results further support the validity of SAXB for the assessment of the X-ray-induced damage through the measurement of the domain dynamics of the crystal. 

Since the SAXB method measures the fluctuation in the domain structure, it could be applicable to measuring and understanding the dynamics of soft materials that have nano- to micro-scale ordered structures such as polymer nanocomposites [17,18], self-assembled lipids [19], porous organic cages [20], and colloidal gels [21].

## 3. Materials and Methods

### 3.1. Preparation of Single Crystals of Lysozyme

Lysozyme from chicken egg white was purchased from Nacalai Tesque Inc. (Kyoto, Japan), and the crystals were prepared in a crystallization solution containing 0.6–1.0 M sodium chloride, 50 mM sodium acetate buffer at pH 4.5, and 25% glycerol at a temperature of 293K.

### 3.2. Measurement of the Time-Resolved X-ray Diffraction

Time-resolved X-ray diffraction measurements were conducted at the beamline BL2S1 of the Aichi Synchrotron Radiation Center (AichiSR). The BL2S1 beamline is sourced from 5T superconducting bending-magnet beam of AichiSR, which is operated in 300 mA top-up injection mode from 1.2 GeV storage ring. BL2S1 optics comprise a vertical-focusing bent-plate mirror and an asymmetric-cut triangular Ge single-crystal monochromator, which is curved for horizontal focusing of the beam onto the sample position. In the DXB experiment, the measurement was performed by loosening the Ge-crystal bend for defocusing to extend the beam area. The X-ray beam of 17 keV was adjusted to the monochromator optics and the 2-theta stage, on which a detector was installed. In addition, the beam was not passed through a collimator to ensure a large irradiation area only through a guiding pipe.

The X-ray diffraction from a lysozyme single-crystal was recorded using a 2D photon-counting detector XSPA-500k (Rigaku, Tokyo, Japan) at a time resolution of 70 μs/frame for 0.7 s. The XSPA-500k is a PAD detector using Si as the sensor material with a pixel size of 76 × 76 μm [22]. This high-speed detector can accept direct X-ray beams without a beam stopper, which enabled us to measure the SAXS region with a short sample-to-detector distance (85 mm). Because of such a short camera distance, the experiment was conducted without a vacuum path.

### 3.3. SAXB Analysis 

SAXB measures the crystal domain from the intensity fluctuation in the SAXS region. One-pixel autocorrelation function (ACF) analysis was performed for the SAXS region near the direct beam (2θ = 0.10–0.88°). Intensity fluctuation in each pixel was analyzed with the following equation:$$I\left(\tau \right)=\frac{<I\left(t\right)\cdot I\left(t+\tau \right)>}{<I{\left(t\right)}^{2}>},$$
where *I*(*t*) represents the diffraction intensity, *τ* is the lag time, and the brackets indicate the time-averaged intensities. The calculated ACFs were fitted to single exponential curves using the following formula: $$ACF\left(t\right)=Aexp\left(-\Gamma t\right)+y,$$
where *A* is the amplitude, *y* is the conversional value, *Γ* is the decay constant, and *t* is the time interval. Parameters *A* and *y* were determined from the calculated ACFs. Parameter Γ was optimized to fit the ACF curve using a nonlinear least squares method. We chose 1-pixel decay constants that satisfy the following conditions: 0  <  *y*, 0  <  A and 0 < *Γ*. These calculations were performed for all target pixels. The averaged ACF and distributions for the decay constants were generated from selected pixels using the above conditions. The distribution of the decay constant values was statistically analyzed with the nonparametric Wilcoxon-rank-sum test, and significant differences were shown as *, **, and *** for *p*-value < 0.05, 0.01, and 0.001, respectively.

## Figures and Tables

**Figure 1 ijms-24-16640-f001:**
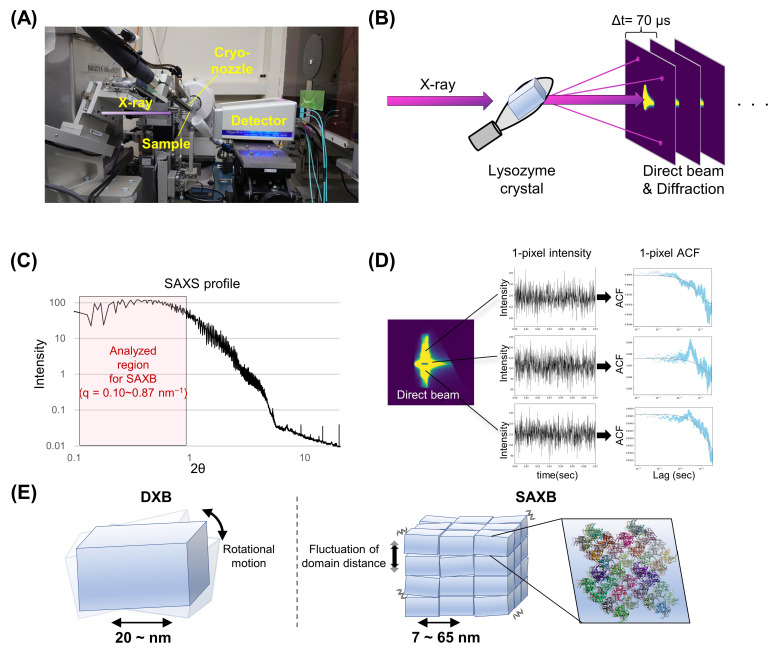
Explanation of the SAXB measurement and analysis. (**A**) Experimental setup for SAXB at BL2S1 of Aichi Synchrotron Radiation Center. (**B**) Schematic illustration of the SAXB measurement. Both X-ray direct beam and diffraction were measured with the XSPA detector at a time resolution of 70 μs. (**C**) Example of converted 1D SAXS profile of a lysozyme crystal. Analyzed region for SAXB (2θ = 0.1~0.9°) was boxed. (**D**) One-pixel ACF analysis for the direct beam. (**E**) Comparison between the target motion of DXB and SAXB. DXB measures the rotational motion of crystalline particles, while SAXB monitors the crystal domain fluctuation.

**Figure 2 ijms-24-16640-f002:**
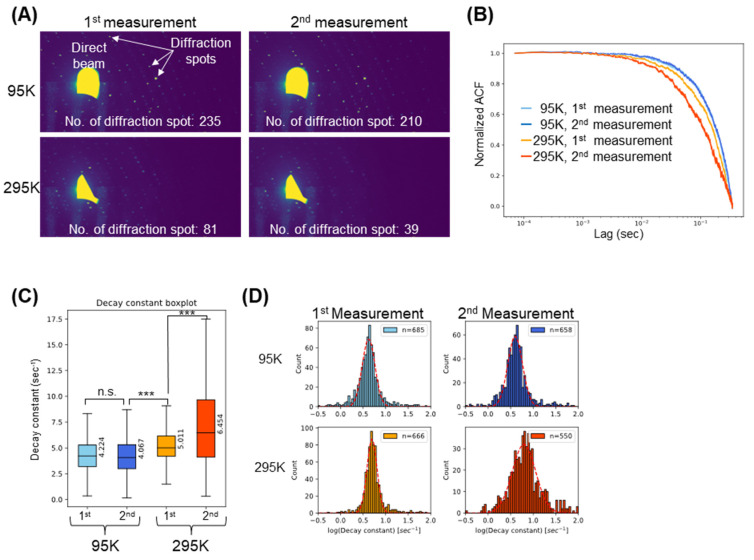
Temperature-dependent SAXB analysis for a lysozyme crystal. (**A**) Time-averaged X-ray diffraction images of a lysozyme crystal at two-different temperatures. Intensity of X-ray diffraction spots dramatically decreased in the 295 K measurement. (**B**) Averaged normalized ACF curves calculated from intensity fluctuation in the direct beam. (**C**) Boxplot of the ACF decay constants calculated from the fitting function against 1-pixel ACF. Median values were shown. n.s. indicates not significant (*p*-value > 0.05), whereas *** indicates significant difference (*p* < 0.001). (**D**) Histogram of logarithm of the ACF decay constant. Red dashed line indicates a Gaussian fitting. The number of pixels used for the analysis was shown on the top right.

**Figure 3 ijms-24-16640-f003:**
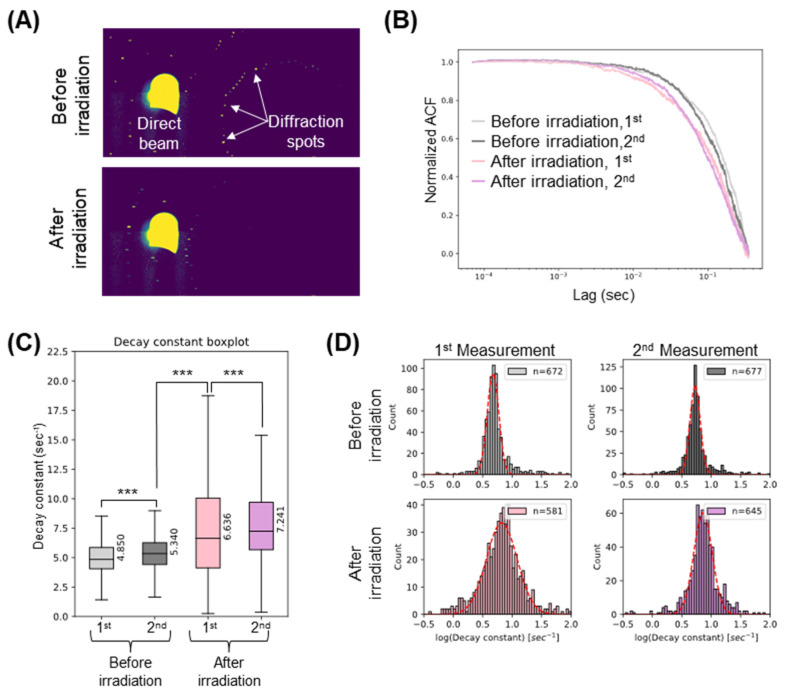
SAXB analysis for lysozyme crystal with/without 100 s X-ray irradiation. (**A**) Time-averaged X-ray diffraction images of a lysozyme crystal before and after 100 s X-ray irradiation. Intensity of X-ray diffraction spots dramatically decreased after irradiation. (**B**) Averaged normalized ACF curves calculated from intensity fluctuation in the direct beam. (**C**) Boxplot of the ACF decay constants calculated from the fitting function against 1-pixel ACF. Median values are shown. n.s. indicates not significant (*p*-value > 0.05), whereas *** indicates significant difference (*p* < 0.001). (**D**) Histogram of logarithm of the ACF decay constant. Red dashed line indicates a Gaussian fitting. The number of pixels used for the analysis is shown on the top right.

## Data Availability

The data presented in this study are available on request from the corresponding author. The data are not publicly available due to their large size.

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
