# Peer review of "The Blinking of Small-Angle X-ray Scattering Reveals the Degradation Process of Protein Crystals at Microsecond Timescale"

_ijms, 2023, doi:10.3390/ijms242316640_

Round 1
Reviewer 1 Report
Comments and Suggestions for Authors The authors have provided a fantastic method to track the protein crystal degradation in real time under X-ray radiation. This method has clearly shown how small angle X-ray blinking, leveraging SAXS and diffracted X-ray blinking can be successfully implemented and demonstrate great promise in the future. The manuscript can be improved after several minor issues are addressed. 1. The data and diagram in Figure 1d are too small and blurry, which is hard to read. 2. Can the authors add a short description of how feasible this method can be applied by other scientists in the world? 3. the authors didn't acknowledge any funding, which is odd. Is this work not funded by any project?Author Response
Thank you for your comments and suggestions. According to your suggestions, we revised the manuscript as follows.
>> 1. The data and diagram in Figure 1d are too small and blurry, which is hard to read.
Page 3, Figure1: Thank you for your suggestion. We have modified Figure 1D in which labels of x- and y-axes of the graphs were enlarged.
On the other hand, since graphs in this figure are just examples of 1-pixel intensity and ACF to show how to analyze, the values are not so informative and their size remained small.
>> 2. Can the authors add a short description of how feasible this method can be applied by other scientists in the world?
Page 5, line 156: We added a short description for the applicability of the SAXB method.
>> 3. the authors didn't acknowledge any funding, which is odd. Is this work not funded by any project?
Page 6, line 216: We added the funding information for this research.
Reviewer 2 Report
Comments and Suggestions for Authors
In this report, the authors examined a combined use of diffracted X-ray blinking (DXB) and small-angle X-ray scattering (SAXS) methods for a lysozyme single crystal to obtain information regarding its domain dynamics caused by X-ray irradiation, which could be correlated with the X-ray induced damage. This small-angle X-ray blinking (SAXB) method would be applicable not only to evaluate the damage of protein crystals but also to visualize the vibration of crystal domain at micro- to milli-second time-scale. This manuscript includes a new methodology of X-ray crystallography, providing valuable insights in the structural biology and biophysics fields. This reviewer would recommend this manuscript for publication in IJMS. However, this reviewer has a minor comment that should be adequately addressed in revision of the manuscript.
1. The authors should more clearly indicate whether same one lysozyme crystal was used for between temperature-dependent (Fig. 2) and dose-dependent (Fig. 3) X-ray damages evaluation measurements. If the authors used only one crystal but not two crystals betwen two experiments, the measurement order should be indicated between two experiments.
2. The authors should explain not only the median values but also the range in the histogram of the decay constant. In the 2nd measurement at “295K” (Fig. 2), the histogram was significantly distributed as compared with those from other three measurements. Moreover, in the 1st measurement “after irradiation” (Fig. 3), the histogram appears to be significantly distributed as compared with the others.
Author Response
Thank you for your comments and suggestions. According to your suggestions, we revised the manuscript as follows.
>> 1. The authors should more clearly indicate whether same one lysozyme crystal was used for between temperature-dependent (Fig. 2) and dose-dependent (Fig. 3) X-ray damages evaluation measurements. If the authors used only one crystal but not two crystals betwen two experiments, the measurement order should be indicated between two experiments.
Page 4, line 137: According to your suggestion, we clearly indicated that we used different crystals for the temperature-dependent (Fig. 2) and dose-dependent (Fig. 3) experiments.
>> 2. The authors should explain not only the median values but also the range in the histogram of the decay constant. In the 2nd measurement at “295K” (Fig. 2), the histogram was significantly distributed as compared with those from other three measurements. Moreover, in the 1st measurement “after irradiation” (Fig. 3), the histogram appears to be significantly distributed as compared with the others.
Page 4, line 123-126 & page 5, 149-153: According to your suggestion, we added descriptions for the distribution of the decay constant value.